# Alarmins and MicroRNAs, a New Axis in the Genesis of Respiratory Diseases: Possible Therapeutic Implications

**DOI:** 10.3390/ijms24021783

**Published:** 2023-01-16

**Authors:** Alessandro Allegra, Giuseppe Murdaca, Luca Gammeri, Roberta Ettari, Sebastiano Gangemi

**Affiliations:** 1Division of Hematology, Department of Human Pathology in Adulthood and Childhood “Gaetano Barresi”, University of Messina, 98125 Messina, Italy; 2Department of Internal Medicine, Ospedale Policlinico San Martino, 16132 Genoa, Italy; 3Department of Clinical and Experimental Medicine, Unit and School of Allergy and Clinical Immunology, University of Messina, 98125 Messina, Italy; 4Department of Chemical, Biological, Pharmaceutical and Environmental Sciences, University of Messina, 98168 Messina, Italy

**Keywords:** respiratory diseases, alarmins, high mobility group box 1, interleukin-33, microRNAs, immune response, inflammation, epigenetics, asthma, acute respiratory distress syndrome

## Abstract

It is well ascertained that airway inflammation has a key role in the genesis of numerous respiratory pathologies, including asthma, chronic obstructive pulmonary disease, and acute respiratory distress syndrome. Pulmonary tissue inflammation and anti-inflammatory responses implicate an intricate relationship between local and infiltrating immune cells and structural pulmonary cells. Alarmins are endogenic proteins discharged after cell injury in the extracellular microenvironment. The purpose of our review is to highlight the alterations in respiratory diseases involving some alarmins, such as high mobility group box 1 (HMGB1) and interleukin (IL)-33, and their inter-relationships and relationships with genetic non-coding material, such as microRNAs. The role played by these alarmins in some pathophysiological processes confirms the existence of an axis composed of HMGB1 and IL-33. These alarmins have been implicated in ferroptosis, the onset of type 2 inflammation and airway alterations. Moreover, both factors can act on non-coding genetic material capable of modifying respiratory function. Finally, we present an outline of alarmins and RNA-based therapeutics that have been proposed to treat respiratory pathologies.

## 1. Introduction

### 1.1. Chronic Respiratory Diseases

Chronic respiratory disorders are the third most important cause of mortality worldwide after cardiovascular pathologies and cancers [1]. Numerous reports show that the occurrence of asthma, chronic obstructive pulmonary disease (COPD), pulmonary tumours, infection and hypertension, acute lung injury (ALI), and acute respiratory distress syndrome (ARDS) is progressively rising [2,3,4].

These diseases are highly heterogeneous, with varying pathogenesis, clinical manifestations or histopathology ranging from slight inflammation to the generation of disseminated nodular lesions or lung fibrosis [5]. However, changes in the humoral or cellular immune responses are described in all subjects suffering from respiratory diseases. These changes can influence the onset, progression, and prognosis of these diseases, and may be therapeutic targets.

The purpose of our review is to highlight the alterations in respiratory diseases affecting specific immunological elements, for example those involving some alarmins such as high mobility group box 1 (HMGB1) and interleukin (IL)-33, and the relationships between them and with genetic non-coding materials, which are unable to determine the production of proteins but can induce significant modifications in gene expression.

### 1.2. General Considerations Regarding High Mobility Group Box 1, Interleukin-33 and microRNAs

#### 1.2.1. High Mobility Group Box 1

HMGB1 is a chromatin-binding protein that weakly binds DNA. It is formed by 215 amino acid residues divided into three different parts: two tandem high mobility group box domains separated by a short linker and an acidic C-terminal tail. It binds DNA to modify the chromatin configuration while preserving genome stability [6].

HMGB1 produces numerous effects inside the nucleus. It helps nucleosome development, operates as a DNA chaperone to support DNA duplication and repair [7,8], and is doubtlessly implicated in the pathogenesis of tumours [9,10]. It is also involved in V(D)J transcription and stimulates autophagy and cell survival when it occurs. It is produced and released by macrophages, natural killer (NK) cells, and dendritic cells (DCs) during inflammation, or by cells after necrosis or cell death.

HMGB1 is an alarmin and, thus, causes reactions of both adaptive and innate immune effectors. It operates via the receptor for advanced glycation end products (RAGE) and other receptors, such as Toll-like receptors (TLRs) 2 and 4 [11]. HMGB1 works as a damage-associated molecular pattern (DAMP) when discharged from injured cells or released from stimulated macrophages, triggering numerous ligands [12,13,14,15,16,17]. This alarmin may be involved in other processes, such as coagulation. HMGB1 intensifies the sterile inflammation correlated with infection and thrombosis [18]. Platelets accumulate HMGB1, transfer it to the cell surface when stimulated and then discharge it into microparticles. Moreover, thrombocytes can connect to HMGB1 via the RAGE, coherent with possible autocrine effects [19,20,21].

#### 1.2.2. Interleukin-33

IL-33 is a 30-KDa protein belonging to the IL-1 cytokine family. Its structure presents two different domains: the C-terminal and the N-terminal domain. The C-terminal domain is responsible for extracellular ST2-dependent activity. The N-terminal domain matches IL-33 to the nucleus and has a transcriptional repressor effect [22]. The IL-33 chromatin-binding motif present in this domain facilitates attachment to histone dimers and changes the chromatin configuration. This mechanism is implicated in the transcription of genes, altering gene repression [8]. The active form of IL-33 does not hold a signal peptide, so it is not discharged via a usual secretory pathway; rather, this cytokine is discharged when cells are injured and works like an alarmin [23]. IL-33 is released by epithelial and endothelial cells but can be released into the airways by other cells, such as DCs and macrophages [24,25].

IL-33 is an essential controller of immune responses operating on DCs, mast cells, group 2 innate lymphoid cells (ILC2s), macrophages, and CD4 + T2 cells, provoking a type 2 immune response [5]. Moreover, this cytokine can stimulate the ST2 of Tregs, demonstrating its capability to reduce inflammation. However, IL-33 can stimulate the synthesis of IL-8 via the stimulation of the JNK/c-Jun/AP-1 pathway, provoking inflammatory conditions. The control of gene expression is also due to the silencing of RNAs. IL-33 has been shown to silence Rip2 by sequentially reducing the expression of mRNAs associated with factors such as IL-17, iNOS, E-selectin, ICAM-1, VCAM-1, and TSLP. Furthermore, NOD-like receptor protein 3 (NLRP3) gene silencing reduces IL-33 synthesis [26].

Finally, it has been reported that IL-33 might also have a double effect in tumour diseases, operating as a pro-cancerogenic or anti-tumorigenic factor according to the cellular setting, expression concentrations, and type of inflammatory context [27,28].

#### 1.2.3. microRNAs

Modifications in gene expression can be caused by numerous forms of non-coding RNA, including long non-coding RNA (lncRNA), circular RNA (circRNA), and microRNA (miRNA). MicroRNAs are RNA sequences of 17–24 nucleotides that modify gene expression by connecting to messenger RNA (mRNA) transcripts in the 3′-untranslated region or coding sequence, causing a reduction in mRNA translation and protein production, and stimulating mRNA elimination [29,30]. However, connecting miRNAs to mRNA transcripts may be an inaccurate process, involving chains of only 6–8 nucleotides; this implies that each miRNA can have several different mRNA targets and, hypothetically, can affect numerous genes and different functional systems [31,32]. There are more than 2000 miRNAs recognized in humans. In the cytoplasm, miRNAs experience the phenomenon of evolution with the Dicer RNAse III endonuclease enzyme. This enzyme determines the introduction of slight alterations in the RNA sequences that can generate sets of similar miRNAs called families; these may or may not possess the same roles and mRNA targets.

Furthermore, miRNAs can be categorized into isomiRs, which contain modifications in the sequence dimension or nucleotides at the 3′ or 5′ ends, occasionally with alterations in activity and targets [33]. It has been shown that although miRNAs are present and operate within the cytoplasm, they can also be discharged into vesicles called exosomes. These vesicles arrive at the circulatory system, where they may facilitate cellular interactions [34]. MiRNA profile changes and altered expression compared to healthy subjects can be observed in different diseases. These variations in miRNA profiles promote some genes and reduce the expression of others [31].

In chronic inflammatory diseases, epigenetic modifications induced by altered miRNA expression cause an imbalance in the homeostasis process. Therefore, these modifications lead to the over-expression of pro-inflammatory cytokines and the suppression of anti-inflammatory factors. Several studies have highlighted the role of miRNAs in the pathogenesis of chronic inflammatory diseases [29].

We found evidence of a correlation between miRNAs and the expression of alarmins; this relationship could allow the modulation of alarmin levels through miRNA mimics or miRNA inhibitors.

In the following sections, we evaluate the role played by HMGB1, IL-33, and non-coding genetic material in the onset of the most important respiratory diseases. In particular, we highlight the presence of any correlations between these factors and the epigenetic role played by these components, even outside their known roles in inflammatory processes.

## 2. Alarmins and Chronic Respiratory Diseases

### 2.1. Asthma and Alarmins

Asthma is a chronic pathology characterized by airway inflammation, hyperresponsiveness, submucosal fibrosis and increased mucus production [35]. It affects more than 270 million people, affecting both children and adults. Approximately 3–10% of all patients present with a severe form of the disease [36].

Asthma is distinguished by several immunological alterations provoked by the relationship between the environment and the patient’s genetic substrate. Based on immunological alterations, the disease can be classified into T2 asthma and non-T2 asthma [37]. This classification depends on the concentrations of type 2 cytokines [38].

The pathogenesis of asthma is due to type 2-mediated allergic inflammation that induces barrier defences at the mucosal surfaces and stimulates the generation of type 2 cytokines, including IL-4, IL-5, IL-9, and IL-13, and the shift of antibodies to IgE [39,40,41]. An innate type 2 immune response requires CD4+ T cells but depends on ILC2s [42,43,44]. ILC2s were initially recognized as cells not derived from the bone marrow and spleen. These are IL-25-dependent non-T non-B cells that were identified in the lung, spleen, liver, mesenteric lymph node, and peritoneum [45,46]. ILC2s induce T-helper cell (Th)2-related lung inflammation by controlling the delivery of Th2 cytokines. IL-33 plus IL-2, IL-7, IL-25, and TSLP stimulate lung ILC2s by reacting to allergen-induced tissue injury [47,48]. Lung ILC2s induce the production of a significant amount of IL-5 and IL-13 but not IL-4. In any case, ILC2 stimulation is mainly due to IL-33, which is induced in type 2 pneumocytes after contact with allergens [49,50,51,52,53]. It is noteworthy that different asthma types may respond to various treatment regimens [54,55].

Several experiments have shown a correlation between asthma, IL-33, non-coding genetic material, and HMGB1. The aggravation of asthma is distinguished by increased inflammation of the airways, which is controlled by IL-33 (Figure 1) that can stimulate lung macrophages. The Th2 cytokine context of allergic asthma may participate in worsening this condition. A study evaluated whether the generation of pro-inflammatory cytokines by IL-33-activated macrophages was increased in cells treated with Th2 cytokines, such as IL-4 and IL-13. In RAW264.7 cells treated with IL-4 and IL-13, the gene expression of *Ccl3, Ccl5, Ccl17, Ccl24,* and *Il1b* after IL-33 activation was increased; this was equivalent to the increased expression of miRNA-155-5p, an miRNA that is expected to control the onset of allergic inflammation. Resolvin (RvE1), a substance that can improve asthmatic inflammation in vivo, reduced the increased generation of *Ccl3*, *Ccl5*, *Ccl24*, and *Il1b* [56]. Therefore, IL-33-stimulated macrophages may participate in airway inflammation and the aggravation of allergic asthma, while RvE1 might operate as a factor that targets macrophages [57,58].

As for HMGB1, some investigations showed that it has a key effect in experimental animal models of type 2 lung diseases. HMGB1 signalling via the RAGE and TLR4 is indispensable for the sensitization of animals to inhaled dust mite allergens. RAGE production by lung cells is essential for provoked airway eosinophilia, the development of ILC2s and the generation of IL-5, IL-13, and IL-33 in a model of eosinophilic airway inflammation [59]. The levels of HMGB1 in the sputum of subjects with severe asthma and of the RAGE and HMGB1 produced by the nasal cells of subjects with chronic rhinosinusitis with nasal polyposis (CRSwNP) and eosinophilic inflammation are higher than in patients with mild disease. This underlines the importance of HMGB1 and the RAGE for respiratory diseases caused by type 2 immunoreactions [60,61,62,63,64].

A different approach to show the correlation between IL-33 and HMGB1 evaluated whether the P2Y_13_ receptor (P2Y_13_-R), a purinergic G protein-coupled receptor (GPCR), controlled the discharge of IL-33 and HMGB1 [65]. Human and mouse airway epithelial cells and C57Bl/6 mice were treated with different aeroallergens or respiratory viruses. The discharge and transfer of alarmins from the nucleus to the cytoplasm were evaluated. The effect of the P2Y_13_-R on the activity of airway epithelial cells was evaluated during and after experimental asthma by employing antagonists or animals with P2Y_13_-R gene deletion. Allergen contact provoked the discharge of ADP and ATP, molecules that stimulate P2Y_13_-R. Allergens, ATP, ADP, or contact with viruses caused the transfer of IL-33 and HMGB1 from the nucleus to the cytoplasm and their subsequent release; this effect was annulled by genetic deletion or the use of P2Y13 antagonists [65]. This work distinguished P2Y_13_-R as a new gatekeeper of the nuclear alarmins IL-33 and HMGB1 and recognized that targeting this GPCR via genetic deletion or therapy with a small-molecule antagonist defended against the onset and exacerbation of experimental asthma.

A different form of asthmatic disease is represented by neutrophilic asthma, which is believed to be a more serious, unrelenting, and steroid-resistant type of asthma that significantly influences the lives of asthmatic patients, with a greater frequency of mortality. Standard therapies are unsuccessful against neutrophilic asthma, and thus, alternative efficient treatments are necessary. Through the study of gene signatures of mast cell activation (on the U-BIOPRED asthma cohort), Tiotiu et al. were able to reconnect distinct transcriptional phenotypes assumed by mast cells and the phenotypes of severe asthma. An IL-33-stimulated mast cell signature was associated with severe neutrophilic asthma [66].

The function of alarmins in some pathological processes, such as ferroptosis, supports the existence of an axis composed of HMGB1 and IL-33. A link between IL-33 and HMGB1 in neutrophilic asthma has been evidenced and could be represented by ferroptosis. Ferroptosis is a type of regulated cell death (RCD) system that is dependent on reactive oxygen species (ROS) [67]. This process is controlled by glutathione peroxidase 4 (GPX4) and is determined by lipid peroxidation due to ROS and iron excess [68]. Ferroptosis is correlated with several lung pathologies, such as lung fibrosis and cancer, COPD, acute pulmonary damage, and eosinophilic asthma [69,70,71,72,73,74,75].

HMGB1 is discharged into the extracellular area in numerous forms of RCD and is correlated to the genesis of inflammation, both due to infections and aseptic. As an alarmin, HMGB1 could provoke the discharge of inflammatory substances, such as cytokines, after ferroptosis initiation. It has been reported that ferroptosis could significantly stimulate inflammation through IL-33 delivery [76]. The reduction in IL-33 decreases the inflammation caused by house dust mites (HDM), including eosinophil and neutrophil infiltration [77].

A different study evaluated whether liproxstatin-1 (Lip-1) modifies the progression of neutrophilic asthma by suppressing ferroptosis and assessed the causal molecular processes [78]. Lipopolysaccharide (LPS) and IL-13 were dispensed to two different bronchial epithelial cells—16HBE and BEAS-2B cells—to produce an experimental model of cell damage. The study employed the ovalbumin (OVA)/LPS-induced animal model.

Lip-1 upturned the modified production of ferroptosis regulators, such as solute carrier family 7 member 11, GPX4, and prostaglandin-endoperoxide synthase 2. On the other hand, Lip-1 reduced lipid ROS and improved cell viability in cells cultured with LPS and IL-13.

Furthermore, Lip-1 administration caused a relevant decrease in the production of IL-33 and HMGB1 in the cultured cells. Finally, the employment of Lip-1 relieved OVA/LPS-induced neutrophilic asthma as suggested by the modification of pathological lung findings and ferroptosis [78]. This study confirmed the significant role of IL-33 and HMGB1 in neutrophilic asthma and suggested that Lip-1 reduced asthma by blocking ferroptosis. This evidence could lead to an innovative approach to neutrophilic asthma therapy.

The main role played by the HMGB1 and IL-33 axis has also been theorised in other experimental models of asthma, such as aspirin-exacerbated respiratory disease (AERD). Liu et al. stated that platelets stimulated via the type 2 cysteinyl leukotriene receptor 2 (CysLT2R) provoked IL-33-dependent pathologic diseases via an inducible system needing the effects of HMGB1 and the RAGE [79]. Leukotriene C4 (LTC4) stimulated surface HMGB1 expression by animal platelets in a CysLT2R-reliant modality. The suppression of the RAGE and inhibition of HMGB1 precluded LTC4-induced platelet stimulation. Tests on AERD-like Ptges−/− mice with inhaled lysine aspirin (Lys-ASA) stimulated LTC4 production and provoked the rapid intrapulmonary accumulation of platelets. With this, a platelet- and CysLT2R-mediated augmentation of IL-33, IL-5, IL-13, and HMGB1 in bronchoalveolar lavage [79] was observed. Therefore, the mechanism of action of CysLT2R is represented by the RAGE/HMGB1 as a connection between type 2 respiratory diseases and the IL-33-dependent mast cell stimulation characteristic of AERD. Employing antagonists of HMGB1 or the RAGE might be advantageous for curing AERD and other diseases correlated with type 2 pathology.

### 2.2. Alarmins and Respiratory Syncytial Virus Infection

Interesting data on the role of HMGB1 and IL-33 have also emerged in the context of different respiratory diseases, such as disorders derived from respiratory syncytial virus (RSV) infection and cystic fibrosis (CF).

RSV is a rapidly transmissible virus that provokes acute lung infections in children. The signs of RSV infection are slight; nevertheless, a subset of patients presents with severe RSV-correlated bronchiolitis. Inflammation provoked by RSV infection can include both the superior zone with nasal inflammation and rhinorrhoea and the inferior respiratory area, resulting in wheezing, bronchiolitis, and coughing [80,81]. Severe RSV-induced bronchiolitis causes necrosis, oedema, the shedding of airway epithelial cells, mucus overproduction and peribronchiolar inflammation. These elements cause airway obstruction [82].

Generally, RSV induces a type 1 immune response. Nevertheless, type 2 cytokines may also be released after RSV infection. There are increasing suggestions that children with critical RSV-correlated bronchiolitis are at a higher risk of presenting with asthma throughout their lives [83,84,85,86].

The epithelial-originated cytokines IL-33 and IL-25 and the alarmin HMGB1 stimulate and control ILC2 activity, driving the development of type 2-provoked pulmonary pathologies [87,88,89].

Clinically, increased quantities of IL-33 have been reported in the nasal fluids of children affected by RSV, connecting IL-33 to the onset of type 2 inflammation after RSV infection. At the same time, pulmonary DCs and alveolar macrophages from the lungs of RSV-infected animals are described to produce IL-33 [90]. The increase in IL-33 expression after RSV infection was verified in vitro by employing the DC cell line DC2.4 and the macrophage cell line RAW264.7. The increased generation of IL-33 was because of TLRs 3 and 7. Moreover, this increase was mediated by raised signalling via the MAPK pathway [91]. Finally, in a different experimental animal model of RSV infection, airway hyperreactivity was caused by IL-33-stimulated ILC2s, leading to IL-13 release [92].

The management of RSV-infected airway epithelial cells in a culture with a xanthine oxidase inhibitor reduced the generation of both IL-33 and TSLP [93]. In vivo, the dispensation of a xanthine oxidase inhibitor reduced the generation of bronchoalveolar lavage fluid (BALF) IL-33 and decreased the number of ILC2s in the lungs of mice [93].

The role played by the IL-33/HMGB1 axis in respiratory infections has been confirmed in other experimental studies. IFN-beta promoter stimulator I (IPS-1) deficit influences susceptibility to viral bronchiolitis and provokes the onset of type 2 inflammation and airway alterations. The main reason for this condition is a deficiency in antiviral cytokine generation that can cause a significantly higher viral burden in epithelial cells, which die due to necrosis, and the production of IL-33 and HMGB1. This condition is associated with the increased generation of ILC2 type 2 cytokines, such as IL-5 and IL-13, and airway smooth muscle modification. Significantly, this type 2 inflammatory condition does not occur in adult animals with an IPS-1 deficit affected by infection, indicating that this condition is dependent on age and reflecting the human distribution of sensitivity to RSV bronchiolitis. The appearance of a type 2 immune response after an acute and severe alteration of epithelial function is coherent with its fundamental function in regulating lesion repair [94,95] as ILC2s, which are the principal ILC component in the lungs, are the main controllers of tissue damage repair [96,97]. For example, the lung alterations occurring after a helminth infestation provoke the discharge of IL-33, increasing the generation of ILC2-originated type 2 cytokines that can re-establish tissue homeostasis. An analogous condition occurs in the genesis of asthma, except that the starting damage is due to an allergen or a respiratory virus [98].

As for HMGB1, some studies reported raised concentrations of this alarmin in the lungs of infected animals compared to normal mice. Meanwhile, compared to normal cells, HMGB1 production in a bronchial epithelial cell line (16HBE) was increased after RSV infection [99]. The administration of glycyrrhizin, a substance able to inhibit HMGB1, after RSV infection decreased the quantity of RSV-infected cells [99].

Finally, different analyses verified the discharge of HMGB1 from RSV-infected airway epithelial cells and showed that ROS production stimulated by RSV infection enhanced the liberation of HMGB1 from infected cells. In addition, they stated that HMGB1 increased the discharge of several cytokines, including IFN, from human monocytes [100,101,102].

### 2.3. Alarmins and Cystic Fibrosis

In CF, recurrent lung inflammation provokes increasing tissue injury. After cell damage and the onset of inflammation, necrotic cells discharge proteins as danger signals, which participate in the eradication of pathogens and stimulate cell repair [103]. However, the excessive liberation of such substances may increase inflammation, accelerating tissue alterations.

Considering the Th17/Th2 form of pulmonary phlogosis in CF patients and its correlation to CF-related pathogens, IL-33 and HMGB1 may participate in inflammation. Unlike the diffuse production of HMGB1, IL-33 is basically produced in epithelial cells [103] and has a role in tissue inflammation correlated to pathogen infection [104,105,106,107].

To evaluate the influence of IL-33 and HMGB1 in CF inflammation, researchers considered the IL-33 concentrations in the BALF of CF patients and evaluated subjects with frequent pulmonary infections. The IL-33 concentrations were significantly higher in CF subjects than in non-CF subjects, while the HMGB1 concentrations were higher in non-CF controls with repeated infections [108]. Furthermore, IL-33 was found to be correlated with IL-8 and IL-13 in CF patients, proving the Th2-increasing and monocyte-recruiting abilities of IL-33 and suggesting a possible effect in epithelial inflammation through IL-8 in CF. The possibility of correlating cytokine production with some clinical aspects of CF appears particularly interesting as, in stable CF subjects, the concentrations of IL-33 in the BALF were negatively correlated with the forced vital capacity. In the same study, the authors evaluated the concentrations of IL-33 in epithelial lung tissue from the explanted end-stage lungs of CF subjects and non-CF patients affected by lung fibrosis and primary pulmonary hypertension and lung fibrosis.

Similarly, according to what has been previously reported, IL-33 expression tended to be greater in CF subjects than non-CF patients, although the study failed to detect a statistical significance [108]. Although it may seem unexpected that HMGB1 is reduced in the BALF of CF subjects compared to non-CF subjects with frequent pulmonary infections, in any case, the HMGB1 concentrations are higher compared to normal subjects. Furthermore, the joining of HMGB1 to bacterial components has been described, and a higher presence of these elements is observed in the lungs of CF subjects compared to non-CF controls. Finally, cholinergic agonists employed in CF treatment inhibit HMGB1 [109]. On the contrary, IL-33 displays more stimulation of the immune system and indicates cell injury.

Furthermore, the Th2 cytokine pattern in CF patients has been correlated to a deficit of IFN-c [110], possibly due to an effect of IL-33. Moreover, IL-33 regulates neutrophil inflow, a specific aspect of CF. In addition, contact with *Pseudomonas aeruginosa* has been reported to stimulate IL-33 but not HMGB1 expression, and this finding might correlate this infection with an increased Th2 reaction [111]. Finally, other relationships between Il-33 and CF have been observed, such as an association between IL-33 expression and the lack of a CF transmembrane conductance regulator [111].

IL-33 might participate significantly in the Th2 reaction reported in CF lungs, facilitating the onset of chronic lung inflammation in CF patients via the inadequate inhibition of inflammatory responses. Therefore, IL-33 may be relevant not only in CF lung pathology but also in other CF-correlated conditions. The different effects of IL-33 and HMGB1 could help create a more precise model of the immune system alterations in CF patients [112].

## 3. MicroRNAs and Chronic Respiratory Diseases

### 3.1. MicroRNAs and Asthma

The role played by miRNAs in respiratory diseases is closely linked to the specific miRNA, as the non-coding genetic material can play a triggering or protective role.

MiRNAs are involved in the genesis of asthma, as a relevant modification in the miRNA generation of airway cells has been described by numerous studies [113,114,115,116,117,118]. An analysis stated that epithelial miRNA-141 controlled the presence of airway mucus in asthma [118], while different studies implicated epithelial miRNAs in airway eosinophilia; moreover, an evaluation of miRNAs in asthma stated that a group of epithelial miRNAs was reduced in these subjects [118]. For instance, miRNA-30a-3pa, an miRNA also implicated in the growth and apoptosis of tumour cells [119,120], was considerably reduced in the blood of asthmatic subjects and is probably involved in airway eosinophilic inflammation. Other studies have confirmed the dysregulation of miRNAs in asthma. MiRNA-218-5p was reported to target δ-catenin and participate in eosinophilic airway inflammation [117]. Similarly, miRNA-145-5p stimulates the appearance of asthmatic attacks through 3A, an inhibitory member of the kinesin family within airway epithelial cells [121]. In contrast, miRNA-23a was reported to play a role in the occurrence of asthma by modifying B-cell lymphoma 2 (BCL2) expression in bronchial epithelial cells and C-X-C motif chemokine ligand 12 (CXCL12) in fibroblasts [122].

Even more interesting are the data relating to a possible axis between miRNAs and HMGB1 in the pathogenesis of asthmatic disease. For instance, miRNA-3934 was studied in the mononuclear cells of asthma subjects and normal subjects [123]: in the study, the cells were administered advanced glycation end products (AGEs) and transfected with miRNA-3934 mimics. The results showed that miRNA-3934 was reduced in the basophils of the asthmatic subjects. At the same time, the levels of pro-inflammatory cytokines, such as IL-33, were increased in these cells. Nevertheless, this finding was overturned by the transfection of miRNA-3934 mimics.

Moreover, it was reported that miRNA-3934 concentrations might be employed to differentiate asthma subjects from non-asthmatic subjects. Regarding the mechanism, miRNA-3934 suppressed the increase in AGEs, which provoked an increase in apoptosis in basophils [123]. These findings suggest that miRNA-3934 can reduce the onset of asthma by targeting the RAGE and probably inhibiting other pathways, such as the TGF-β/Smad signalling pathway.

Similarly, the increased expression of miRNA-126 reduced lung endothelium damage and raised survival in endotoxin-induced animals [124] and decreased endothelial barrier injury by controlling the production of tight junction proteins including claudin, zonula occludens-1 and occludin [125,126]. However, a different mechanism has been reported, and experimentation revealed that miRNA-126-3p carries out its protective function on endothelial barrier integrity through its effect on Phosphoinositide-3-Kinase Regulatory Subunit 2 (PIK3R2) with the stimulation of the Akt pathway [127].

A particularly interesting report investigated the RNA profile (mRNA, miRNA and lncRNA) in T2 asthma via airway biopsies [128]. The results showed that 202 mRNAs, 22 miRNAs and 30 lncRNAs were differentially present in biopsies from T2 asthma subjects. Among them, the lncRNA PCAT19 presented great diagnostic accuracy in differentiating T2 asthma subjects from non-T2 asthma subjects and non-asthmatic subjects. Moreover, it is known that RNAs can regulate each other functionally, and a competing endogenous RNA (ceRNA) complex was identified involving eight mRNAs, 12 miRNAs and 13 lncRNAs [128]. The results of this experiment may help to discover new therapeutic strategies for T2 asthma patients. In this condition, PCAT19-dominated ceRNA regulation systems may have a crucial effect, and PCAT19 may be useful as a possible immune-correlated marker for asthma or other respiratory pathologies associated with eosinophilic inflammation.

The data relating to another group of miRNAs are also interesting, namely the miRNA-17~92 group, which is an intensely powerful controller of B and T cell growth, maturation and stimulation, and cytokine generation [129]. This set of miRNAs is transcribed as a single miRNA transcript modified by the Drosha/Dgcr8 microprocessor complex and Dicer to generate six different miRNAs. These miRNAs can be categorized into different families: the miRNA-17 -18, -19, and -92 families. MiRNA-19a is increased in airway-penetrating T cells from asthmatic patients and stimulates Th2 cytokine generation via a direct effect on several signalling inhibitors, such as suppressor of cytokine signalling 1 (SOCS1), PTEN, and TNFAIP3, which encodes A20.

In other studies, authors have evaluated the miRNA transcriptomes of lung Th2 cells and ILC2s. The expression of the miRNA-17~92 cluster was extremely relevant to preserving the ILC2 equilibrium in vivo. Furthermore, miRNA-17~92 stimulated IL-5 and IL-13 liberation after IL-33 and ILC2-determined type 2 inflammation in vivo [130]. After contact with the allergen papain, animals lacking the miRNA-17~92 groups in ILC2s presented with decreased lung inflammation. Moreover, miRNA-17~92-deficient ILC2s showed irregular proliferation and cytokine production after the administration of IL-33 and thymic stromal lymphopoietin in vitro. miRNA-19a stimulated IL-13 and IL-5 generation and suppressed the production of numerous targets, including A20 and SOCS1, inhibitors that reduced IL-13 and IL-5 generation. These results confirm the role of miRNAs as essential controllers of ILC2 activity and support the possible therapeutic activity of targeting miRNA-19 to reduce allergic responses [131].

Even more interesting are the data relating to a possible correlation between miRNAs and HMGB1 in the pathogenesis of asthmatic disease. There are three runt-related transcriptional factor (RUNX) genes, namely *RUNX1*, *RUNX2,* and *RUNX3*, as well as maternal smoking, which have been reported to potentially support the onset of asthma in children by increasing the expression of *RUNX1* [132]. Moreover, RUNX2 is described as stimulating the gene transcription of a sterile alpha motif (SAM) pointed domain comprising an ETS transcription factor (*SPDEF*) and may join the promoter of *HMGB1* [133,134]. Wu et al. evaluated miRNA-30a-3p production in asthma patients and non-asthmatic subjects and considered the relationship between miRNA-30a-3p and airway eosinophilia [135]. They found that miRNA-30a-3p production was significantly reduced in the bronchial brushings of asthma patients compared to normal controls. Epithelial miRNA-30a-3p production was negatively related to several factors indicating airway eosinophilia, such as eosinophils in sputum or bronchial biopsies and exhaled nitric oxide in patients. The authors demonstrated that *RUNX2* is a target of miRNA-30a-3p and augments HMGB1 expression. HMGB1 and RUNX2 production are both increased in the airway epithelium and are related to each other in asthmatic subjects. A reduction in miRNA-30a-3p increased RUNX2 and HMGB1 production, while increasing RUNX2 stimulated HMGB1 in BEAS-2B cells. Interestingly, an airway increases in mmu-miRNA-30a-3p inhibited HMGB1 and RUNX2 expression and reduced airway eosinophilia in an experimental animal model [135]. Therefore, epithelial miRNA-30a-3p could target the RUNX2/HMGB1 axis to reduce airway eosinophilia in asthma (Figure 2).

### 3.2. MicroRNAs and Acute Respiratory Distress Syndrome

ALI or its more critical manifestation, ARDS, is a severe acute pulmonary pathology with a significant mortality rate. ALI/ARDS is caused by different pulmonary conditions, such as pneumonia, or indirect extra-pulmonary damage, such as sepsis [136]. Its characteristics include alteration of the alveolar-capillary membrane, increased inflammation and reduced alveolar fluid clearance, with secondary hypoxemia, pulmonary oedema, and abnormal gas exchange [137,138,139].

ARDS occurs in about 10% of subjects entering an intensive care division, with a mortality rate oscillating from 35% to 46%. Currently, no standard pharmacological approaches exist for this condition [140,141].

MiRNAs and HMGB1 could also play a role in the genesis of ARDS. A report stated that the increased expression of miRNA-181b reduced the gene expression of importin-a3 and was able to decrease lung damage and mortality rates in ARDS animals [142]. Furthermore, Rao et al. showed that modification of the cytokine signalling suppressor 1 (SOCS1) in an animal model of ARDS reduced inflammatory cytokine production and inflammatory cell accumulation in miRNA-155 (−/−) animals compared to wild-type mice [143].

Finally, in different experiments, researchers assessed which miRNA could improve ARDS by targeting HMGB1 [144]. In experimental in vitro and in vivo models of LPS-induced ARDS models, they studied the effect of miRNA-574-5p on the production of HMGB1, pro-inflammatory cytokines and inflammasomes. MiRNA-574-5p seemed able to reduce the inflammatory reaction by operating on HMGB1. Stimulating the production of miRNA-574-5p or HMGB1 siRNA silencing reduced the stimulation of the NLRP3 inflammasome. Additionally, increased expression of HMGB1 overturned the anti-inflammatory action of miRNA-574-5p. In vivo, the increased production of miRNA-574-5p reduced interstitial oedema and alveolar leucocyte infiltration in ARDS animals [144]. This study may offer new therapeutic approaches for ARDS.

## 4. Exosomal MicroRNA, Circular RNA, and Long Non-Coding RNA in Respiratory Diseases

Not only circulating miRNAs but also those contained in exosomes could play a role in respiratory diseases. Endothelial progenitor cells (EPCs) are an encouraging possibility for ALI treatment [145,146]. Some studies have demonstrated that EPCs can reduce inflammation and vascular leakage and increase bacterial elimination in ALI and infection-induced pulmonary damage [147,148,149,150]. Lately, exosomes have appeared as a relevant paracrine system that allows cell-to-cell communication by easing the transport of miRNAs from one cell to another [151]. For instance, it was previously demonstrated that EPC exosomes could transport miRNA-126 into endothelial cells, causing a decrease in the activation of genes significant for the onset of ARDS, while fibroblast (NIH3T3) exosomes (which do not include a relevant amount of miRNA-126) possess slight protective activity [152]. Moreover, the employment of EPC exosomes reduced infection-correlated ALI in animal models, an effect caused by miRNA-126. Exosomes can be collected in a longstanding modality and present several advantages compared to cell-based treatment [153].

Other studies confirmed that EPC exosomes could have positive effects in ALI [154]. The intratracheal dispensation of EPC exosomes decreased pulmonary damage after LPS-provoked ALI at 24 and 48 h and led to a reduction in protein and chemokine levels in the BALF. Moreover, this treatment reduced the pulmonary injury score and oedema, suggesting a decrease in inflammation and permeability. Furthermore, in small airway epithelial cells, the increased expression of miRNA-126-3p can modify phosphoinositide-3-kinase regulatory subunit 2, while the increased expression of miRNA-126-5p inhibits HMGB1 and the permeability element Vascular-Endothelial Growth Factor-alpha (VEGFα). Significantly, both miRNA-126-3p and 5p increase the production of tight junction proteins, indicating a possible process by which miRNA-126 may reduce LPS-provoked pulmonary damage [154].

Other forms of non-coding genetic material could function as mediators in respiratory diseases. circRNAs are a group of covalently closed RNAs generated from the ‘back-splicing’ of primary transcripts that are more stable compared to linear RNAs [155,156,157].

The significant alteration of circRNAs has been demonstrated in respiratory diseases. For instance, circ_0007385 presence is greater in non-small cell lung cancer (NSCLC) cells and is correlated with a poor prognosis. Silencing circ_0007385 is possible to reduce cell growth and invasion in A549 and H1975 cells and decrease cisplatin resistance [158]. Furthermore, circ_0007385 silencing reduced the tumour proliferation of A549 cells in vivo. There was a direct interface between HMGB1, miRNA-519d-3p and circ_0007385. The production of miRNA-519d-3p was reduced in NSCLC in a circ_0007385-related manner, and circ_0007385 could secondarily control HMGB1 through miRNA-519d-3p. Thus, by both reducing miRNA-519d-3p and re-establishing HMGB1, it is possible to reverse the inhibitory action of circ_0007385 reduction in cell growth and invasion [158].

Finally, lncRNAs can influence numerous biological processes due to their ability to bind DNA, RNA, and proteins. The altered expression of lncRNAs may play a role in various diseases; Gál et al. aimed to evaluate their expression in chronic respiratory diseases. The differential expression of lncRNAs in asthma and COPD highlights their role as biomarkers; it also implicates them in the pathomechanism of these diseases [159].

## 5. Discussion

Evidence suggests that a relevant role is played by alarmins and miRNAs in the determinism of respiratory diseases. IL-33 and HMGB1 have a significant role in Th2 inflammation and severe asthma [56,61]. In addition, alarmin levels correlate negatively with pulmonary function [62,63] and promote the remodelling of airways and exacerbation of disease [77]. Furthermore, these alarmins seem capable of modifying the activity of immunological effectors, favouring the appearance of an immune profile capable of promoting both infectious and other respiratory pathologies. IL-33 stimulates lung natural helper cells to produce IL-13 [91]. In addition, IL-33 activates ILC2s and increases bronchial responsiveness [92]. In RSV infection, HMGB1 stimulates epithelial cells and monocytes [100]. Finally, Il-33 and HMGB1 contribute to lung inflammation in CF patients [105,108,112].

Furthermore, the close link between alarmins and non-coding genetic material appears particularly interesting. Many studies have demonstrated the potential utility of miRNAs as biomarkers for numerous diseases. Furthermore, the study of miRNAs involved in respiratory diseases and their role in the regulation of alarmins can be significant for new therapeutic strategies.

Several miRNA/alarmin axes have been identified. Chia et al. observed an increase in IL-33 associated with an increase in miR-155 in macrophages cultured with IL-4 and IL-13. Future studies are necessary to clarify the link between alarmin and miR-155 [56].

MiR-3934 appears to have a protective effect on the respiratory system, suppressing the release of IL-33 and other pro-inflammatory cytokines through the inhibition of the RAGE and, consequently, of the TGF-β/Smad pathway [123].

Moreover, the miR-17~92 cluster is involved in ILC2 homeostasis. MiR-17∼92-deficient ILC2s showed defective cytokine expression when stimulated by IL-33 [131].

Another interesting axis that has emerged is between miR-30a-3p, RUNX2, and HMGB1. In a mouse model of asthma, the overexpression of miR-30a-3p suppressed RUNX2 and HMGB1 and decreased eosinophilic inflammation [135].

In ARDS models, miR-574-5p appears to suppress inflammation by targeting HMGB1 [144].

Zhou et al. found a link between miRNA-126-5p and HMGB1 in mouse models of LPS-induced ALI [154]. Blocking these axes upstream could lead to the development of more effective therapeutic strategies than those currently known. These could also be used in severe diseases that do not respond to conventional therapy.

Protective ventilation is the most effective treatment against ALI/ARDS, which comprises excellent positive end-expiratory pressure values and small tidal volumes [160]. However, mortality persists at an elevated level (about 40%) as the long-term use of mechanical ventilation can worsen pulmonary injury [161].

The inhibition of pathways that intensify inflammation related to this critical condition might be an interesting strategy for reducing pulmonary damage [162]. Gene silencing may be the best way to block these systems and represents a new therapy for ALI/ARDS. In particular, gene silencing through short-interfering RNA (siRNA) may be used to change the production of the pro-inflammatory substances in ALI/ARDS [163].

Several pre-clinical studies demonstrate the appropriateness of siRNA-based treatments for ALI/ARDS by regaining the normality of pulmonary endothelial and epithelial defences. Therapeutic approaches, such as the use of a combination of siRNAs inhibiting diverse mRNAs, might be employed in a sequence for aiming at multiple targets to offer a synergistic silencing result against the numerous pathways implicated in ALI/ARDS. Moreover, the co-transport of siRNA with additional therapeutic factors might improve the therapeutic factors. New nebulization devices are also required to enhance the delivery of siRNA/carrier to the most distal parts of the lung [164].

IL-33 itself could be a target for asthma therapy, and different studies evaluated the effects of hydrogen gas on asthma and tried to clarify the underlying molecular processes [165,166]. Experiments showed that molecular hydrogen decreased IL-33 production in HDM-stimulated human bronchial epithelial cells (16HBE) at the transcriptional and post-transcriptional levels [167]. HDM significantly increased the protein and mRNA concentrations of IL-33 in 16HBE cells, when hydrogen gas was suppressed. These modifications were due to the change in the transcription activity as HDM intensely increased the IL-33 promoter effect. Hydrogen gas significantly reduced the HDM-provoked increase in the expression of IL-33, suggesting that molecular hydrogen not only operated as an antioxidant [168,169], but also regulated IL-33 production by controlling its transcription in HDM-stimulated 16HBE cells. Meanwhile, the IL-33 receptor ST2 displayed a similar profile to IL-33 after HDM and hydrogen administration, suggesting that hydrogen reduced the auto-intensification of the IL-33/ST2 axis [170].

HDM exposure was also capable of changing the miRNA profile. Soldberg et al. stated that 21 miRNAs were increased, and 19 miRNAs were decreased by HDM in the bronchial epithelial cell line 16HBE. Among them, miRNA-1246 was highly increased by HDM exposure by more than 3.2-fold [171].

In a different study, miRNA-21-3p was increased in 16HBE cells by HDM, and this rise was eliminated by hydrogen gas. MiRNA-21 was reported to cause severe, steroid-resistant allergic airway pathologies by targeting the phosphatase and tensin homolog to control PI3K signalling [172]. Nevertheless, further studies must clarify how these miRNAs are controlled by molecular hydrogen and allergens.

Hydrogen gas reduced BALF and serum IL-33, IL-4, IL-25, and TSLP concentrations as well as lineage−ICOS+ ST2+ ILC2s in asthmatic animals, suggesting that hydrogen alleviated OVA-induced type II inflammation via targeted IL-33/ST2 signalling [170]. Finally, examining the miRNAs involved in the various pathogenic mechanisms is essential. For example, the imbalanced regulation of bronchial inflammatory cell survival and apoptosis plays a key role in asthma. Baculoviral IAP repeat-containing 5 (BIRC5), known as urviving, could be an important regulator of asthmatic processes, probably due to the inhibition of eosinophilic apoptosis. Identifying the epigenetic factors involved in urviving regulation could lead to new approaches to controlling the disease [173].

The tables below summarize the studies on the role of alarmins (Table 1) and miRNAs (Table 2) in the main respiratory diseases.

## 6. Conclusions

Great focus has been put on the immune system as the principal element capable of inducing pulmonary pathologies, such as asthma, ARSD, CF, and RSV infection. In these pathological conditions, the effects of alarmins have a significant impact, and the immunological changes they cause are responsible for the correlated phlogistic status. However, new routes have started emerging in the field of research on the genesis and treatment of respiratory pathologies. The notion that non-hematopoietic tissues are a fundamental font of chemokines has been sustained by several experiments [174].

The airway epithelium is not just the first barrier against inspired allergens but has a relevant immunomodulatory function in the lung [175]. Increasing findings demonstrate that these cells are the functional contributors to controlling the immune response to environmental factors through the discharge of elements, such as IL-33, and DAMPs, such as HMGB1. Current knowledge considers the epithelium of airways as an innate immune tissue that recognizes inspired allergens via an arsenal of receptors and triggers innate and adaptive immunity, certainly also via the effect of alarmins. IL-33 and HMGB1 are pro-inflammatory factors that can be secreted by bronchial epithelial cells and could drive the release of inflammatory cytokines in response to several stimuli. The blockade of this functional axis could suppress inflammation, alleviate inflammatory cell infiltration, goblet cell hyperplasia and airway mucus production in the lungs, and reduce cellular damage.

However, it is possible that the role of alarmins does not end in the induction of a specific immune reaction but also occurs through the intervention of epigenetic modifications. There are numerous suggestions of pulmonary pathologies and environmental contacts correlating with specific methylation patterns in blood cells and epithelial or lung fibroblasts. However, the biological functions of these methylation modifications and the clinical consequences for diagnosis, prognosis and treatment are inadequately recognized and investigated [176]. The role of non-coding RNAs in the pathogenesis of chronic lung diseases has been observed, and there is evidence of the existence of connections between miRNAs and alarmins. In addition, several in vitro and in vivo (murine models) studies have demonstrated how the modulation of specific miRNAs can regulate the expression of pro-inflammatory cytokines and alarmins.

The use of target therapies that block specific miRNAs (antagomiRs) or miRNA analogues (miRNA mimics) could be a promising approach for severe chronic respiratory diseases in the future via the inhibition or modulation of alarmin production or delivery. There is evidence in the literature of the importance of studying the alarmin/miRNA axis (Table 3), but some analysed studies have several limitations. First, many clinical studies have been performed on small cohorts of patients. Moreover, the application of mRNAs as biomarkers of disease or disease severity is still difficult in the clinic as not all laboratories own the necessary tools. Finally, several studies on mouse models or in vitro have given excellent results in terms of therapeutic perspectives, but further investigations are required for their application in humans.

Regardless of the evident relationship between miRNAs and epigenetics, studies on the impact of IL-33 on immune effectors have begun to unveil different directing pathways at the plane of cellular activities and epigenetic modifications. In reaction to the IL-33 stimulus, numerous functional systems are interlinked with transcriptional programs, therefore governing the destiny of immune cells. Ultimately, it is important to clarify the intricate function of IL-33 in lung diseases, driving the elaboration of innovative approaches for immune treatment [177]. Similarly, HMGB proteins are implicated in structural modifications caused by chromatin remodelling elements. Specific attention should be given to the DNA chaperone system, in which HMGB proteins insert bends into the double helix, making the DNA reachable for effector proteins and facilitating their action [178].

All these new data further clarify the mechanisms that regulate the onset and progression of respiratory diseases and will allow the development of adequate therapeutic approaches for these pathological conditions.

## Figures and Tables

**Figure 1 ijms-24-01783-f001:**
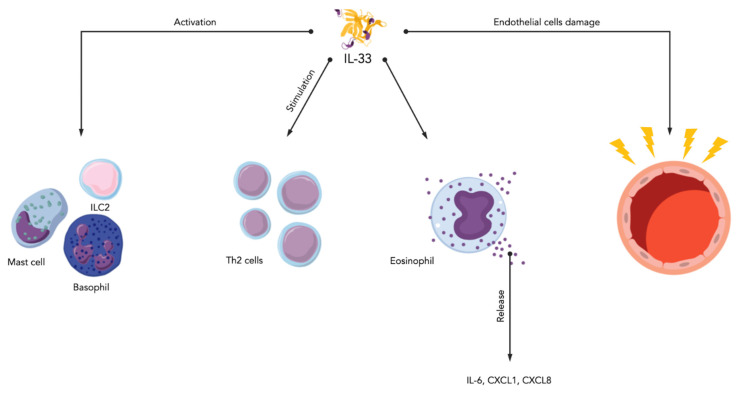
Involvement of IL-33 in asthma. IL-33 stimulates cells of the immune system (ILC2s, mast cells, Th2 cells, eosinophils, and basophils) to release pro-inflammatory cytokines and chemokines. IL-33 also causes endothelial cell damage.

**Figure 2 ijms-24-01783-f002:**
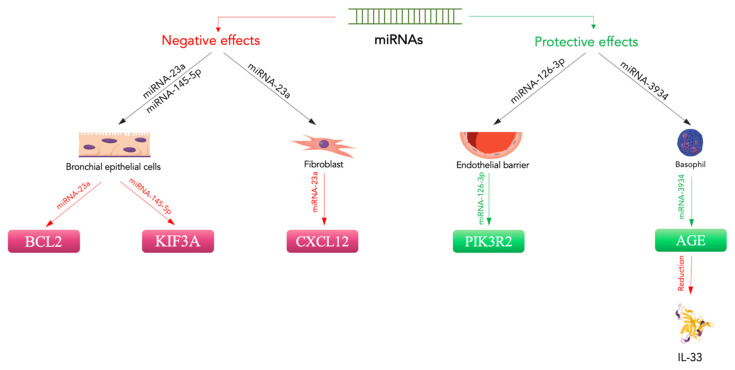
Protective and negative effects of miRNAs on asthma. MiRNA-23a downregulates CXCL12 expression in fibroblasts [122]. In bronchial epithelial cells, it negatively regulates BLC2 expression, while miRNA-145-5p suppresses KIF3A [121,122]. These miRNAs promote bronchial inflammation and disease exacerbation. In contrast, protective effects are related to the action of miRNA-126-3p on the endothelial barrier through PIK3R2 [127]. Additionally, miRNA-3934 acts to mitigate IL-33 levels [123]. Magenta indicates downregulation, green indicates upregulation.

**Table 1 ijms-24-01783-t001:** Effects of alarmins on respiratory diseases.

Authors	Chronic Respiratory Disease	Alarmins	Purpose of the Study	Model of Study	Results
Chia et al. [56] (2018)	Asthma	IL-33	To demonstrate that IL-33-activated macrophages contribute to the worsening of airway inflammation in exacerbations of allergic asthma	In vitro	IL-33-stimulated cells led to an increase in pro-inflammatory cytokines potentially involved in allergic asthma exacerbations.
Watanabe et al. [60] (2011)	Asthma	HMGB1	To demonstrate the correlation between HMGB1, neutrophilic inflammation and disease severity in asthma	Human	HMGB1 levels in induced sputum were significantly higher in asthmatic patients than in healthy controls and were inversely correlated with %FEV1 and FEV1/FVC. HMGB1 levels were directly proportional to disease severity and the percentage of neutrophils in induced sputum.
Shim et al. [61] (2012)	Asthma	HMGB1	To evaluate the role of HMGB1 in the pathogenesis of eosinophilic asthma	Human and mouse	Sputum HMGB1 levels were increased in asthmatics compared with controls and positively correlated with the number of sputum eosinophils and the expression of TNF-α, IL-5, and IL-13. In a mouse model of asthma, eosinophilic airway inflammation was attenuated by the inhibition of HMGB1.
Cuppari et al. [62] (2015)	Asthma	HMGB1	To evaluate HMGB1 levels in sputum in children with stable allergic asthma and their correlation with lung function	Human	Sputum HMGB1 levels were increased in children with asthma compared with healthy controls. HMGB1 correlated positively with total IgE levels and negatively with lung function.
Hong et al. [63] (2013)	Chronic rhinosinusitis	HMGB1	To measure the expression of HMGB1 in the paranasal sinus mucosa and the difference in expression between CRS patients and healthy controls	Human	By both RT-PCR and real-time PCR, HMGB1 mRNA expression was significantly increased in the tissues of CRS patients compared with controls. HMGB1 protein level was significantly increased in CRS tissues by Western blot. Immunohistochemistry demonstrated that HMGB1 was mainly expressed in CRS.
Dzaman et al. [64] (2015)	CRSwNP	HMGB1	To evaluate the expression of the RAGE and HMGB1 in CRSwNP and their correlation with the risk of relapse and disease severity	Human	The expression levels of the RAGE and HMGB1 in tissues correlated with disease severity. Elevated RAGE expression was associated with increased disease severity, allergy, and AERD in patients with CRSwNP.
Werder et al. [65] (2022)	Asthma	HMGB1 and IL-33	To evaluate whether P2Y13-R (P2Y13 receptor) regulates the release of IL-33 and HMGB1	Mouse (in vivo and in vitro)	In vitro exposure to aeroallergens or viruses induced the release of IL-33 and HMGB1. This response was blocked by the genetic deletion or pharmacological antagonism of P2Y13. In mouse models, the prophylactic or therapeutic isolation of P2Y13-R attenuated the onset of asthma and reduced the severity of rhinovirus-associated exacerbation.
Allinne et al. [77] (2019)	Asthma	IL-33	To demonstrate that increasing IL-33 levels maintains pulmonary inflammation, promoting remodelling and exacerbation	Mouse	IL-33 drives inflammation and bronchial remodelling. The use of an IL-33 neutralizing antibody reduced inflammation and improved the remodelling of both lung epithelium and lung parenchyma.
Liu et al. [79] (2019)	AERD (aspirin-exacerbated respiratory disease)	HMGB1	To demonstrate that platelets activated through CysLT2R facilitate the genesis of IL-33-dependent eosinophilic inflammation through the action of HMGB1 and the RAGE	Mouse	Mouse models subjected to platelet depletion, HMGB1 neutralisation and RAGE isolation did not respond to lysine-aspirin (Lys-ASA) inhalation. They also did not record an increase in IL-33, mast cell activation and changes in airway resistance.
Saravia et al. [90] (2015)	Respiratory syncytial virus infection	IL-33	To demonstrate that IL-33 release in the lungs during RSV infection is age-dependent	Mouse	RSV infection induced IL-33 expression and ILC2 increase in the lungs of neonatal mice but not adults. Using anti-IL-33 antibodies or an IL-33 receptor knockout mouse during infection inhibited the immunopathogenesis of RSV.
Liu et al. [91] (2015)	Respiratory syncytial virus infection	IL-33	To investigate the role of natural lung helper cells in the infiltration of eosinophils into the lung using BALB/c mice infected with RSV	Mouse	RSV infection induced an increase in the absolute number of natural helper cells in the lungs of mice and an increase in IL-13, suggesting that these cells may be the source of IL-13. Lung natural helper cells produced IL-13 stimulated by IL-33, which was especially increased in the lungs of mice after intranasal RSV infection.
Wu et al. [92] (2020)	Respiratory syncytial virus infection	IL-33	To study the role in the pathophysiology triggered by RSV infection of each innate immune cell activated by IL-33 (including ILC2s and ST2+ myeloid cells)	Mouse	IL-33-activated ILC2s led to the development of airway hyperresponsiveness (AHR) and inflammation during RSV infection. Myeloid cell-derived IL-33 was required for airway inflammation, suggesting the importance of IL-33 signalling.
Fonseca et al. [93] (2020)	Respiratory syncytial virus infection	IL-33	To evaluate the role of the uric acid pathway during RSV infection	Mouse (in vivo and in vitro)	The inhibition of uric acid pathway activation during infection reduced the expression of IL-33, TSLP and CCL2 in airway epithelial cells and IL-1β in bone marrow-derived macrophages. A reduction in ILC2s, macrophages, and IL-33 was observed in mice treated with XOI or interleukin-1 receptor antagonist during infection.
Manti et al. [99] (2018)	Respiratory syncytial virus infection	HMGB1	To evaluate the role of HMGB1 in RSV infection	Mouse (in vivo and in vitro)	RSV infection strongly induced HMGB1 expression both in vitro and in vivo.
Hosakote et al. [100] (2016)	Respiratory syncytial virus infection	HMGB1	To study the mechanism of action of HMGB1 released extracellularly in airway epithelial cells to establish its role in RSV infection	In vitro	RSV infection of human airway epithelial cells induced a significant release of HMGB1. Treatment with antioxidants considerably inhibited the extracellular release of HMGB1. HMGB1 appears to function as a paracrine factor by stimulating epithelial cells and monocytes.
Rayavara et al. [101] (2018)	Respiratory syncytial virus infection	HMGB1	To study the mechanism of action of HMGB1 released extracellularly in airway epithelial cells to establish its role in RSV infection	In vitro	HMGB1 determined the phosphorylation of NF-κB and P38 MAPK but did not stimulate cytokine release from airway epithelial cells. HMGB1 induced the release of cytokines from immune cells.
Norlander et al. [102] (2020)	Respiratory syncytial virus infection	HMGB1 and IL-33	To describe the contributions of ILC2s and alarmins in RSV infection	Review	From the literature review, it emerged that ILC2s and alarmins are key mediators in the early phase of the type 2 response to RSV infection.
Roussel et al. [105] (2013)	Cystic fibrosis	IL-33	To demonstrate the correlation between IL-33, CFTR mutations and CF-related lung diseases	In vitro	IL-33 was upregulated in CF-related lung disease. IL-33 was present in the AEC nuclei of patients with CF and was released after tissue injury. IL-33 bioactivity was increased by neutrophil elastase.
Rowe et al. [108] (2008)	Cystic fibrosis	HMGB1	To determine whether HMGB1 contributes to lung inflammation in cystic fibrosis and regulates neutrophil chemotaxis and lung matrix degradation	Human and mouse	Sputum HMGB1 levels were increased in subjects with CF. The chemotaxis of human neutrophils stimulated by purified HMGB1 was partially dependent on CXC chemokine receptors. The intratracheal administration of purified HMGB1 induced the recruitment of neutrophils to the airways of mice.
Tiringer et al. [112] (2014)	Cystic fibrosis	HMGB1 and IL-33	To evaluate the expression of IL-33 and HMGB1 in the lungs of patients with stable cystic fibrosis	Human	IL-33 levels were significantly higher in clinically stable CF patients than in healthy subjects. HMGB1 levels were higher in non-CF controls with recurrent infections. In clinically stable CF patients who were able to perform pulmonary function tests, IL-33 levels were negatively correlated with forced vital capacity.
Zhang et al. [167] (2021)	Asthma	IL-33	To evaluate the effects of molecular hydrogen on the pathogenesis of asthma	Mouse (in vivo and in vitro)	The serum and BALF levels of IL-33 were significantly increased by OVA and inhibited by H^2^ in mice. H^2^ reduced the HDM-induced apoptosis of 16HBE cells and upregulation of IL-33.
Magat et al. [170] (2020)	Asthma	IL-33	To evaluate the role of IL-33 and its self-amplification of the IL-33/ST2 pathway in Ag-dependent and Ag-independent asthma-like models	Mouse	IL-33 auto-amplified itself and ST2 protein expression in airway epithelial cells. IL-33/ST2 pathway auto-amplification was not present in IL-33 knockout mice. In ST2 knockout mice, IL-33-induced eosinophilic airway inflammation was completely decreased.

**Table 2 ijms-24-01783-t002:** Effects of miRNAs on respiratory diseases.

Authors	Chronic Respiratory Disease	MiRNA	Purpose of the Study	Model of Study	Results
Heffler et al. [113] (2017)	Asthma	Main miRNAs involved in the pathology	To evaluate the main miRNAs involved in asthma and their potential roles as biomarkers or therapeutic targets	Review	Several miRNAs are involved in asthma and could be disease biomarkers or therapeutic targets.
Bartel et al. [114] (2018)	Asthma	MiR-142-3p	To study the connection between miR-142-3p and the proliferation of airway smooth muscle (ASM) precursor cells in asthma	Human and mouse	MiR-142-3p was increased in hyperproliferative regions of the lung in asthma. Bronchial biopsies of patients with early or late-onset severe asthma showed differential expression of miR-142-3p.
Pua et al. [115] (2019)	Asthma	MiR-223, miR-142a	To study the ex-miRNA release during inflammation	Mouse	The extracellular microRNAs (ex-miRNAs) present in the lung had a composition related to the epithelial lining of the airways.
Zhang et al. [116] (2018)	Asthma	MiR-221-3p	To study the correlation between epithelial and sputum miR-221-3p and eosinophilic inflammation in asthma and to evaluate miR-221-3p as a new biomarker	Human	After induction of allergic airway inflammation, epithelial and sputum miR-221-3p were found to be excellent biomarkers of eosinophilic airway inflammation in asthma. Decreased epithelial miR-221-3p led to the upregulation of the anti-inflammatory chemokine CXCL17.
Liang et al. [117] (2020)	Asthma	MiR-218-5p	To determine the role of epithelial miR-218-5p and its target gene in eosinophilic airway inflammation	Human and mouse	The epithelial expression of miR-218-5p was significantly reduced in asthmatic patients and negatively correlated with eosinophils and other type 2 biomarkers. CTNND2 (encoding δ-catenin) was a target of miR-218-5p and its epithelial expression was positively correlated with airway eosinophilia.
Huo et al. [118] (2016)	Asthma	MiR-181b-5p	To evaluate the association between miR-181b-5p and eosinophilic airway inflammation and the possible mechanism by which miR-181b-5p participates in eosinophilic inflammation	Human (in vivo and in vitro)	The epithelial expression of miR-181b-5p was reduced in subjects with asthma and inversely correlated with sputum and bronchial submucosal eosinophilia. Plasma miR-181b-5p increased after treatment with inhaled corticosteroids. In human bronchial epithelial cells, miR-181b-5p regulated IL-13-induced IL-1β and CCL11 expression by targeting SPP1.
Xiong et al. [121] (2019)	Asthma	MiR-145-5p	To study the mechanisms by which miR-145-5p can induce asthma	Mouse	In the airway epithelial cells of asthmatic mice exposed to house dust mites, KIF3A was increased, while miR-145-5p was decreased. The use of miR-145-5p antagonists significantly improved the symptoms.
Jin et al. [122] (2019)	Asthma	MiR-23a	To evaluate the effect of miR-23a on BCL2 and CXCL12 in asthma	Mouse	MiR-23a was upregulated in lung tissues after exposure to the antigen. BCL2 in the epithelium and CXCL12 in fibroblasts were downregulated. The use of a miR-23a mimic or inhibitor changed the expression of BCL2 and CXCL12.
Dou et al. [123] (2022)	Asthma	MiR-3934	To study the role of miR-3934 in the pathogenesis of asthmatic disease	Human (in vivo and in vitro)	MiR-3934 was downregulated in the basophils of asthmatic patients. The use of miR-3934 mimics resulted in a reduction in the expression of the IL-6, IL-8, and IL-33 pro-inflammatory cytokines. MiR-3934 proved to be a good biomarker.
Wang et al. [128] (2022)	Asthma	Main miRNAs involved in the pathology	To identify the competing endogenous RNA network mechanism underlying T2 asthma	In vitro	A total of 30 lncRNAs, 22 miRNAs and 202 mRNAs were differentially expressed in airway biopsies from patients with T2 asthma.
Simpson et al. [130] (2014)	Asthma	MiR-19a	To study miRNAs and their pathways that control the responses of type 2 helper T cells (Th2 cells) involved in the pathogenesis of asthma	Human and mouse	Elevated expression of miR-19a was revealed in human airway-infiltrating T cells from patients with asthma. Modulation of miR-19 activity weakened Th2 cytokine production in both human and mouse T cells.
Singh et al. [131] (2017)	Asthma	MiR-19a	To evaluate the role of miRNAs in the regulation of ILC2 biological activity	Mouse (in vivo and in vitro)	Papain-exposed mice without the miR-17~92 cluster showed reduced airway inflammation. The miR-17~92 cluster member miR-19a induced IL-13 and IL-5 production through the inhibition of SOCS1 and A20.
Wu et al. [135] (2022)	Asthma	MiR-30a-3p	To study the correlation between miR-30a-3p, RUNX2, and HMGB1 in allergic airway inflammation	Human and mouse	The expression of miR-30a-3p was significantly reduced in asthmatic patients compared with control subjects. The epithelial expression of miR-30a-3p was negatively correlated with eosinophils in induced sputum and bronchial biopsies and the fraction of exhaled nitric oxide in patients with asthma. RUNX2 is a target of miR-30a-3p, and the airway overexpression of mmu-miR-30a-3p suppressed the expression of RUNX2 and HMGB1, relieving airway eosinophilia.
Sun et al. [142] (2012)	ARDS	MiR-181b	To evaluate the role of miR-181b in NF-κB-mediated EC activation and vascular inflammation in response to pro-inflammatory stimuli	In vivo and in vitro	MiR-181b inhibited the expression of importin-α3- and NF-κB-sensitive genes, such as adhesion molecules VCAM-1 and E-selectin. The inhibition of miR-181b exacerbated endotoxin-induced NF-κB activity, leucocyte influx and lung injury. Critical patients with sepsis showed reduced levels of miR-181b compared with control ICU subjects.
Rao et al. [143] (2014)	ARDS	MiR-155	To study miRNAs overexpressed after exposure to staphylococcal enterotoxin B (SEB) and the consequent development of acute inflammatory lung injury	Mouse	The most expressed miRNA was miR-155; miR-155(−/−) mice were protected from SEB-mediated inflammation and lung injury. There was a functional link between SEB-induced miR-155 and IFN-γ. MiR-155(−/−) mice showed increased expression of SOCS1, and miR-155 overexpression led to its suppression and a reduction in IFN-γ.
He et al. [144] (2021)	ARDS	MiR-574-5p	To study the correlation between miR-574-5p and HMGB1 and possible new therapeutic strategies for the treatment of ARDS	Human and mouse (in vivo and in vitro)	The expression of miR-574-5p was upregulated in ARDS patients and after LPS stimulation in vitro and in vivo; miR-574-5p suppresses inflammatory responses and NLRP3 inflammasome activation through the inhibition of HMGB1.
Zhou et al. [154] (2019)	ARDS	MiR-126	To study the effects of endothelial progenitor cell (EPC) exosomes and miR-126 in LPS-induced acute lung injury (ALI)	Mouse	The intratracheal administration of EPC exosomes reduced lung injury after LPS-induced ALI. Furthermore, EPC exosomes reduced myeloperoxidase (MPO) activity, lung injury score and pulmonary oedema.
Solberg et al. [171] (2012)	Asthma	MiR-34/449 family	To determine if airway epithelial miRNA expression is impaired in asthma and identify IL-13-regulated miRNAs	Human	Compared with control subjects, 217 miRNAs were differentially expressed in subjects with steroid-naïve asthma and 200 miRNAs in subjects with steroid-using asthma. Treatment with inhaled corticosteroids induced a statistically significant change for only nine miRNAs.
Kim et al. [172] (2017)	Asthma	MiR-21	To study mouse models with severe and steroid-insensitive asthma to identify pathogenic mechanisms and study new therapeutic approaches	Mouse	Infection induced increases in miR-21 expression levels in the lung during SSIAAD, while expression of the miR-21 target phosphatase and tensin homolog was decreased. Treatment with a miR-21-specific antagomir (Ant-21) increased tensin and phosphatase homolog levels. This led to the suppression of airway hyperresponsiveness and the restoration of steroid sensitivity to allergic airway disease.

**Table 3 ijms-24-01783-t003:** Key points.

Key Points
Specific miRNAs can be useful biomarkers of disease and indicate its severity. MiRNAs can also be therapeutic targets or miRNA mimetics can be used to treat respiratory diseases. There are alarmin/miRNA axes. Specific alarmin/miRNA axes can be studied in order to better understand the pathogenetic mechanisms underlying respiratory diseases. The study of alarmin/miRNA axes can lead to the development of new therapeutic strategies.

## Data Availability

Not applicable.

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
