# Peer review of "Alarmins and MicroRNAs, a New Axis in the Genesis of Respiratory Diseases: Possible Therapeutic Implications"

_ijms, 2023, doi:10.3390/ijms24021783_

Round 1

Reviewer 1 Report

This article reviews the progress of research on alarmins (mainly HMGB1 and IL-33) and miRNAs in respiratory diseases.Asthma, cystic fibrosis, respiratory syncytial virus infections, and ARDS/ALI are diseases with significant heterogeneity, and the lack of effective treatments is a major clinical challenge.If we can find the exact target from both alarmin and non-coding RNA, it will be very beneficial for the treatment of these diseases. We can find the authors' efforts in this area, and it would be a very interesting work if the following two problems can be solved well:

1.The authors review the research progress of alarmin and miRNA in certain respiratory diseases, respectively, but can targeted and solid research evidence be provided about the role of a precise alarmin-miRNA axis in these diseases?

2.The authors review alarmin mainly in relation to diseases such as asthma, respiratory syncytial virus infection, and cystic fibrosis, and later mention ARDS/ALI in their review of miRNA findings, and later also in relation to exosomal miRNA and circulatory RNA.Has the author considered the possibility of optimizing and adjusting the framework structure of the text to enhance readability?

Author Response

Dear reviwer, many thanks for your suggestions. Please see attached response point to point

Reviewer 2 Report

The manuscript from Allegra and colleagues deals with a potentially promising topic, the role of alarmins and microRNA in respiratory diseases.

the authors have done a massive work of searching the literature and have collected a considerable amount of information, but there are some concerns:

There are a many typos in the text, an accurate rereading is advisable before resubmission

There are words that have a different meaning and should be corrected:

-abstract, line 22 and throughout the manuscript:"reparation" should "repair"

- line 173 and throughout the manuscript: "grave" means tomb. the correct word is "severe"

-line 45: when writing the extended form of an acronym, please use all first letters as capital (i.e. High Mobility Group Box) or none of them (i.e. high mobility group box)

- line 61: "comprise" should be "including"?

-line 62 and line 464: senteces repeated

-line 142 T4+ cells should be T CD4+ cells

-line 161: genes names should be written in italics, protein names in capital letters

-line 187 define AEC

- line 239 "testes" is a typo

-line 374 a C is missing in CXC motif

Table1 and 2 are poorly informative in the way they are presented; I don't see the point of indicating the country in which the study was conducted, even more if they deals with in vitro or in vivo experiments. It would be more informative if the authors would have indicated if the study cited were in vitro, in vivo or human studies. Moreover, instead of presenting the aim of the studies presented in the table, it would be more helpful to indicate the results of the studies.

In table 2, the header of column #4 claims "alaramins", but there are only RNA listed; alarmins are proteins or peptides.

Figure legends are missing: there is only the figure's titles, but a description would be helpuful and informative. In figure 2 there is a typo (nevative instead of negative).

Author Response

Dear reviewer many thanks for your suggestions, please see attached response point to point

Reviewer 3 Report

This is atimely and comprehensive review on an important field of severe asthma. 

The authors are to be commended for their systematic evaluation of the current state of the art and citing wide scale of relevant publications. 

It would be useful to frame  in a bit wider scope and include BIRC5 (survivin) and other important elements as well in the pathomechanism os eo or non eo asthma. Group of C Szalai also described lncRNAs in asthma. 

Regarding this, please add relevant results from major severe asthma studies such as the  SARP, ADEPT, and U-BIOPRED in Europe,  such as  Tiotiu A, et al on  Association of Differential Mast Cell Activation with Granulocytic Inflammation in Severe Asthma. Am J Respir Crit Care Med. 2022 or others as they handle data obtained from large number of patients.

Also please elaborate on the potential limitations of small scale studies and if possible on the reproducibility of data in different settings.

Author Response

(The authors gave the same response as above.)

Round 2

Reviewer 1 Report

No

Author Response

Dear Referee, you can see the response to your and academic editor suggestions, Giuseppe Murdaca

Reviewer 2 Report

the authors replied to my concerns

Author Response

Dear Referee, you can see the response  academic editor suggestions,

Giuseppe Murdaca
